# Identification of the Human Papillomavirus Genotypes, According to the Human Immunodeficiency Virus Status in a Cohort of Women from Maputo, Mozambique

**DOI:** 10.3390/v14010024

**Published:** 2021-12-23

**Authors:** Cremildo Maueia, Alltalents Murahwa, Alice Manjate, Soren Andersson, Jahit Sacarlal, Darlene Kenga, Tufária Mussá, Anna-Lise Williamson

**Affiliations:** 1Division of Medical Virology, Department of Pathology, Faculty of Health Sciences, University of Cape Town, Cape Town 7925, South Africa; alltalents.murahwa@uct.ac.za (A.M.); anna-lise.williamson@uct.ac.za (A.-L.W.); 2Departamento de Microbiologia, Faculdade de Medicina, Universidade Eduardo Mondlane, Maputo P.O. Box 257, Mozambique; alimanjate28@gmail.com (A.M.); jahityash2002@gmail.com (J.S.); darlene.bintikenga@gmail.com (D.K.); tufariamussa@gmail.com (T.M.); 3Instituto Nacional de Saúde, Vila de Marracuene, Maputo 3943, Mozambique; 4School of Medical Sciences, Örebro University, 702 81 Örebro, Sweden; soren.andersson@folkhalsomyndigheten.se; 5Unit for Vaccination Programs, Public Health Agency of Sweden, 171 65 Solna, Sweden; 6Institute of Infectious Disease and Molecular Medicine and Division of Medical Virology, Department of Pathology, Faculty of Health Sciences, University of Cape Town, Cape Town 7925, South Africa; 7SAMRC Gynaecological Cancer Research Centre, Faculty of Health Sciences, University of Cape Town, Cape Town 7925, South Africa

**Keywords:** human papillomavirus, human immunodeficiency virus, women

## Abstract

Background: Human papillomavirus (HPV) infection is now a well-established cause of cervical cancer and other anogenital cancers. An association between human immunodeficiency virus (HIV) infection and higher HPV incidence and prevalence are commonly reported. This study was conducted to demonstrate HPV prevalence, genotypes and its characteristics, according to the HIV status in women from Maputo in Mozambique. Methods: A total of 233 participants with ages ranging from fourteen to forty-five were included. Cervical samples were collected, DNA extracted, and HPV genotyping was performed using the HPV Direct Flow CHIP Kit. Results: In total, 177 HIV-negative and 56 HIV-positive women were included in the analysis. The overall HPV prevalence was 63% and was significantly higher among HIV-positive women (79% versus 58% among HIV-negative women; *p* = 0.005). The prevalence of multiple HPV type infections was 32%. High-risk HPV types 52, 68, 35, 18 and 16 were the most frequent. A higher proportion of HIV-positive women had multiple HPV types compared with HIV-negative women. Conclusions: This study demonstrated a high prevalence of HPV in the study cohort. HIV-positive women were identified as having the highest HPV prevalence and infection with multiple HPV types across all ages. High-risk genotypes were the most commonly found.

## 1. Introduction

Human papillomavirus (HPV) infection is now a well-established cause of cervical cancer and there is growing evidence of HPV being a relevant factor in other anogenital cancers (anus, vulva, vagina and penis) as well as head and neck cancers [1]. Cervical cancer (CC) is the third most common cancer among women worldwide and in 2018, there were an estimated 569,847 new cases and 311,365 deaths related to CC [2]. Africa has an estimated population of 372.2 million women aged 15 years and older who are at risk of developing cervical cancer. Current estimates indicate that every year 119,284 women are diagnosed with cervical cancer and 81,687 die from the disease [3]. In addition, in Africa, CC ranks as the second most frequent cancer among women after breast cancer [3]. The World Health Organization (WHO) strategy seeks to meet several targets by 2030 for the elimination of cervical cancer, such as: (1) complete vaccination (with the HPV vaccine) of 90% of girls by the age of 15 years; (2) screening (with a high-performance test) of 70% of women (first by the age of 35 years, and then by 45 years), and (3) treatment of 90% of women diagnosed with cervical disease (including 90% of women with cervical pre-cancer and 90% of women with invasive malignancy). According to the WHO, this strategy highlights that investment in the interventions to meet these goals can generate economic and societal returns [4].

Both HIV and HPV are sexually transmitted viruses [5]. Since HPV is more infectious than HIV, the prevalence of HPV is significantly higher among women and men with high-risk sexual behaviour [2,6]. HIV-positive individuals are more likely to have multiple HPV infections as well as higher viral load compared to their HIV-negative counterparts [7,8]. HPV-associated cancers occur more frequently in HIV-positive than in HIV-negative individuals [7].

The HPV types 16, 18, 26, 31, 33, 35, 39, 45, 51, 52, 53, 56, 58, 59, 66, 68, 73 and 82 are causally associated with vulvar, vaginal, cervical, anal, penile and oropharyngeal cancers [1] and are classified as high-risk genotypes considering they association with disease in these organs [3,9]. The HPV types 16 and 18 are responsible for about 70% of all cervical cancer cases worldwide [1,10] and more than 75% of other cancers [1]. Three HPV vaccines are currently in use worldwide, namely, bivalent vaccine (Cervarix^®^, GlaxoSmithKline) targeting HPV 16 and 18, the quadrivalent vaccine (Gardasil^®^, Merck), targeting HPV 6, 11, 16, 18 and nonavalent vaccine (Merck) which protects against quadrivalent vaccine types and 5 additional genotypes (HPV 31, 33, 52, 56 and 58) [11]. These vaccines can prevent girls and boys from acquiring HPV types that cause the majority cervical cancers and other HPV associated cancers and the Merck vaccines prevent 90% of genital warts [12,13]. HPV vaccines that prevent HPV 16 and 18 infections are now available in Africa and have the potential to reduce the incidence of cervical and other anogenital cancers [14].

The East African region is one of the most affected with an age standardized incidence rate of 42.7 per 100,000 women [3]. In this region, Mozambique has the second highest incidence of CC after Malawi. In Mozambique, a population of 8 million women aged 15 years and older are at risk of developing cervical cancer and estimates indicate that every year more than 5600 women are diagnosed with CC of whom 4061 die from the disease [15]. Cervical cancer ranks as one of the most frequent cancer among women in Mozambique and is the most frequent cancer among women between 15 and 44 years of age. About 8.4% of women in the general population are estimated to harbour cervical HPV 16 and 18 infections at a given time, and 51.0% of invasive cervical cancers are attributed to HPVs 16 or 18 [3,15,16]. In some studies, performed on young women in Mozambique indicate that HPV 52 is the most frequent type found in women, followed by HPV 6, 16, 35, 51, 53, and 58 [16,17]. In a study performed on cervical vaginal samples from students in Maputo, HPV16 was the most frequent genotype, followed by HPV58, HPV66, HPV52, HPV18, HPV56, HPV61, and HPV70 [15,16].

HIV infection and sexual debut before 18 years of age are factors associated with multiple HPV infections. HPV vaccine has not yet been introduced in the national expanded programme of immunisation in Mozambique. Taking into consideration the WHO global initiative to accelerate the elimination of cervical cancer by pursuing three important steps, namely: vaccination, screening and treatment, as well as the improve substantially cervical cancer outcome through early detection and effective treatment in low-income countries. To provide insights to inform the strategy for elimination of cervical cancer in Mozambique, the aims of this study was to determine the prevalence of HPV types and its characteristics stratified by age and HIV status, in women recruited from a Health Center (Maputo).

## 2. Materials and Methods

### 2.1. Study Design and Specimen Collection

This cross-sectional study was conducted on non-pregnant women seeking care regarding gynaecological symptoms such as: venereal pain, genital ulcers and vaginal discharge or even family planning in health facilities belonging to Mavalane Health’s area in Maputo, between February 2018 and July 2019. Exclusion criteria included pregnancy, current use of antibiotics or other antimicrobial medications (except antiretroviral medications), menstruation at the time of visit and vaginal douching during the last 7 days.

The study followed the tenets of the Declaration of Helsinki of 2013. Written informed consent was sought from the participants. The Mozambican National Bioethics for Health Committee (CNBS) (ref: 423/CNBS/2018) and the Human Research Ethics Committees of the University of Cape Town (UCT) (HREC reference 850/2019), approved all aspects of the study. Sociodemographic data and information on risk factors were obtained through interviews.

A total of 233 participants aged between 18 and 45 years old were recruited and the objectives of the study explained by members of the research team. After obtaining informed written consent, participants were interviewed twice using a semi-structured questionnaire regarding socio-demographical information, sexual behaviour and genital symptoms by different members of the research team in order to double-check answers. Pre-HIV testing counselling and an HIV-rapid test were performed in all women with unknown HIV serostatus, using the national serial testing algorithm which employs two sequential HIV-1/2 rapid tests: the Determine HIV 1/2 test (Alere Abbott Laboratories, Tokyo, Japan), used for screening and UniGold HIV test (Trinity Biotech, Wicklow, Ireland), used to confirm initial reactivity on the Determine assay. Indeterminate samples were repeated, using fourth generation test Enzygnost ELISA Anti-HIV 1/2 Plus (DADE-Behring, Marburg, Germany).

After a speculum examination, cervical samples were collected by rotating a cytobrush at an angle of approximately 360° at the bottom of the posterior vaginal section. The brushes were placed in a test tube containing 10 mL of a SurePath liquid-based cytology (Thin Prep, Becton, Dickinson and Company, Franklin Lakes, NJ, USA) and stored at −80 °C. Two mL of this samples was transferred to the University of Cape Town (UCT) and used for deoxynucleotide acid (DNA) extraction and HPV genotyping.

### 2.2. DNA Extraction and HPV Genotyping

A total volume of 2 mL Thin Prep cervical specimen was centrifuged at 8000× *g* for 30 min at 4 °C, and cell pellet was resuspended to 400 µL phosphate-buffer saline. DNA from resuspended cells was extracted by a MagNA Pure Compact (Roche Diagnostics, Indiana, USA) using the MagNA Pure Compact Nucleic Acid Isolation Kit (Roche Diagnostics, Indiana, USA) following the manufacturer’s instructions. HPV genotyping was performed using the HPV Direct Flow CHIP Kit (Vitro Master Diagnóstica, Sevilla, Spain) which allows the qualitative detection of 35 types of HPV (high-risk HPV 16, 18, 26, 31, 33, 35, 39, 45, 51, 52, 53, 56, 58, 59, 66, 68, 73 and 82, and low-risk HPV 6, 11, 40, 42, 43, 44, 54, 55, 61, 62, 67, 69, 70, 71, 72, 81 and 84) by amplification of a fragment in the viral region L1 of papillomavirus by PCR, followed by hybridization onto a membrane with DNA-specific probes by using the DNA-Flow technology for manual hybriSpot platforms (Vitro Master Diagnóstica, Sevilla, Spain). The biotinylated amplicons generated after the PCR were hybridized in membranes containing an array of specific probes in a three-dimensional porous environment for each target as well as amplification and hybridization control probes. Once the binding between the specific amplicons and their corresponding probes has occurred, the signal was visualized by an immunoenzymatic colorimetric reaction with Streptavidin−Phosphatase and a chromogen (NBT-BCIP) generating insoluble precipitates in the membrane in those positions in which there has been hybridization. The results were analysed automatically with the HybriSoft software (Vitro Master Diagnóstica, Sevilla, Spain).

### 2.3. Statistical Analyses

The data obtained through the questionnaire was checked for accuracy and double-entered into the Microsoft Excel (2016) computer programme. Exploratory analysis was first carried out to obtain descriptive statistics. Charts and tables were used to summarize data and display figures where appropriate. Data were analysed using Stata 14. HPV infection(s) were compared between HIV-negative and HIV-positive women, overall and stratified by age, using χ2 tests or Fisher’s exact tests in the case of sparse data. Trends in the prevalence of HPV by age were examined using non-parametric tests for trends across age groups. Single HPV infection was defined as infection where only one HPV type was detected, while multiple infection was defined by the presence of two or more HPV types. HPV was further categorised into high-risk (HR), low-risk (LR), or both high- and low-risk according to the types detected. Differences were considered to be statistically significant when *p*-values were <0.05.

## 3. Results

### 3.1. Demographic and Clinical Data of Study Participants

A total of 177 HIV-negative and 56 HIV-positive women were included in the analysis, with the HIV-positive women being significantly older (median age: 34 years versus 22 years among HIV-negative women; *p* < 0.001; Table 1). The majority of participants reported that their sexual debut occurred between the ages of 16–18 years, and most reported having no new sexual partners during the past 3 months. Vaginal discharge was commonly identified (92% among HIV-negative women versus 86% among HIV-positive women), and vaginal ulcers were more common among HIV-positive women (57% versus 44% among HIV-negative women; *p* = 0.088).

### 3.2. HPV Prevalence according to Age and HIV Status

Table 2 presents the prevalence of HPV stratified by HIV-status and age. The overall HPV prevalence in this sample was 63% and was significantly higher among HIV-positive women (79% versus 58% among HIV-negative women; *p* = 0.005). This difference was most apparent among women aged 36–45 years: in this age group, the prevalence of HPV was 80% among HIV-positive women compared to 27% among HIV-negative women (*p* = 0.001). Among HIV-negative women, the HPV prevalence decreased with increasing age (*p* = 0.035). This scenario was not observed among HIV-positive women (*p* = 0.663).

The overall prevalence of multiple HPV type infections was 32% but significantly higher among HIV-positive women (50% versus 27% among HIV-negative women; *p* = 0.001). These differences were most pronounced in the age groups 14–25 and 36–45 years. The prevalence of single HPV infection was 28% and did not differ by HIV-status (*p* = 0.832) or in any age groups.

In the total sample, 28% of women had high-risk HPV only, 13% had low-risk HPV only, and 19% had both high- and low-risk types. The prevalence of high-risk only and low-risk only did not differ by HIV-status, but HIV-positive women were significantly more likely to have both high- and low-risk types (32% versus 15% among HIV-negative women; *p* = 0.005).

No association was found between HPV and any of vaginal ulcer (*p* = 0.269), vaginal discharge (*p* = 0.852), or cervical inflammation (*p* = 0.426).

### 3.3. Prevalence and Distribution of Single and Multiple HPV Genotype Infections

Among the high-risk HPV types, HPV 52, 68, 35, 18 and 16 were the most frequent with 20 (8.6%), 18 (7.7%), 17 (7.3%), 15 (6.4%) and 14 (6%) cases respectively. HPV-58 and 18 were the most frequent among HIV-positive women (Figure 1A). Among low-risk HPV genotypes, HPV 62/81 and 44/55 were the most frequent with 20 (8.6%) and 14 (6%) cases respectively and HPV 44/55, 54 and 62/81 were the most frequent in HIV-positive women (Figure 1B). Figure 2A shows the distribution of multiple infections in HIV-positive and HIV-negative women. In HIV positive women, the higher proportion of detected HPV types was composed by multiple infections than single infection. Both high- and low-risk HPV genotypes together in HIV-positive women was observed in higher frequency (Figure 2B) while high-risk only infections were observed in same frequencies in HIV-positive and negative women.

In Figure 3 is shown the distribution of high and low-risk genotypes in for both HIV-positive women and HIV-negative women. In high-risk genotypes infections, was observed a higher proportion of single infections for both HIV positive and negative women while for low-risk genotypes infections, singles infections were observed in negative women. In HIV-positive women, the infections are generally characterised by more than one infection.

## 4. Discussion

The study demonstrates a high HPV prevalence in this population (63%) which was significantly higher among HIV-positive women 79% versus 58% among HIV-negative women (*p* = 0.005); with a notably higher prevalence of multiple concurrent infections in the HIV-positive group. Several previous studies from the Sub-Saharan Africa region, identified a strong positive association between HIV and HPV infection [3,18]. In our study cohort, HPV prevalence decreased with age and the similar scenario has been reported in other studies [18]. However, when grouped according to HIV-status, the HPV prevalence decreased with age only among HIV-negative women. Similar results were previously reported in a South African and Brazilian studies [5,19]. While several studies have shown a high prevalence of HPV in young women this generally is not the case in older women. In this study the HPV prevalence in HIV negative women was 69.2% (18/26) in 26–35-year-old group which is higher than usually observed in this age group. Mbulawa et al. reported a prevalence of 39% in this age group [19]. Studies in sex workers, conducted by Bui et al., (2018) and Cameron et al., (2018), showed that HPV prevalence declined with age [9,20]. In our study, the high prevalence probably reflects an increased risk of HPV infection due to the cohort being recruited from people seeking consultations for possible sexually transmitted infections (STIs). This may also indicate that they are at risk of new HPV infections due to high risk-behaviours or the high-risk behaviour of their partners.

It is notable in this study that the overall HPV prevalence and multiple infection prevalence remained high (over 78%) in HIV positive women across the age groups of 14 to 25 years and 36 to 45 years while it generally decreased with age among HIV-negative women. In a study conducted by McDonald et al., (2014) similar results were seen among South African HIV-positive women [21] and they stated that the high rate of HPV reactivation may be a result of a suppressed immune system. They also speculated that susceptibility to new infections in HIV-positive women could be the cause of the higher prevalence of HPV in older HIV-positive women. Some authors suggested that immune senescence is the reason that older HIV positive women are more likely to fail to clear HPV infection they acquired at a young age or later [22]. Thus, a more intensive cervical screening program is needed due to the high HPV prevalence and multiple infections across all age group among HIV-positive women [23,24].

It is not biologically clear whether multiple concurrent HPV infections increase the risk of cervical cancer [10,25]. Some research has demonstrated an increased risk of carcinogenesis due to multiple concurrent infections as well as the association with a higher rate of persistent HPV infection and the development of HPV-related cancers compared to infection with a single HPV genotype [26]. In addition, infection with multiple HPV types was previously reported to complicate the response to treatment [27]. Beyazit et al., (2018) studied association between multiple concurrent infections and the risk of presence of Pap smear abnormality and found a significantly positive association in HIV-infected women compared to their HIV-uninfected counterparts [28,29]. Furthermore, when looking only at HIV-infected women, a large association with multiple concurrent HPV infections and abnormal cytology was found, indicating a possible additive effect of HIV infections and infection with multiple concurrent HPVs [29]. However, the impact of multiple concurrent HPV infections on the duration of infection is still uncertain as some research has found no impact of multiple concurrent infections on HPV persistence [5,30,31].

In our study, the most frequent HR-HPV types were HPV 52, 68, 35, 18 and 16. This is in partial agreement with other studies conducted in the same region. In a study conducted by Edna Omar et al. in Mozambican healthy young women, it was demonstrated that HR-HPV genotype distribution in Mozambican healthy young women differs from the global figures with HPV52, 35, 16, 53, 58 and 51 being the most prevalent [16]. Castell-sagué et al., described previously that HR-HPVs 51, 35, 18, 31 and 52 were the most commonly found in women aged 14–61 years with normal cytology in a study conducted in cohort of women from the general population with and without invasive cervical cancer, in southern Mozambique [32].

In the sub-Saharan African region, the most frequently detected HPV types in women with single and multiple HPV infections and normal cytology, were HPV 16 (50.7%), HPV 18 (19.2%), HPV 45 (10.1%), HPV 35 (9.7%), HPV 33 (5.0%), and HPV 52 (4.5%) [10]. In a study performed on self-collected cervical vaginal samples from students in Maputo HPV16 was the most frequent genotype, followed by HPV58, HPV66, HPV52, HPV18, HPV56, HPV61, and HPV70 [17]. In our study population, HPV 16 was not the most prevalent genotype, which differs from worldwide studies. HPV studies conducted in sub-Saharan Africa, indicate that in this region women are significantly less likely to be infected with HPV 16 than their counterparts in Europe and South America and are significantly more likely to be infected with high-risk genotypes other than HPV 16 [18]. The genotypes HPV 35, 45, 52, 56 and 58 are all significantly more common in HPV-positive women in sub-Saharan Africa than in Europe with HPV 35 being the most notable [33].

In the Eastern Africa regions where Mozambique is located, HPV-16 and HPV 52 are found being the most prevalent genotypes in women with cervical cancer as well as in the HIV co-infected women [34,35]. Although HPV 16 and HPV 18 are the two mostly prevalent HPVs, in Sub-Saharan African HPV 45 and HPV 35 are the two additional important types also frequently reported [10,36]. HPV 16, 18, 35, 45, 52 and 58 were also identified as common types in an African woman with cytological abnormalities as well as cervical cancer [33]. Thus, the presence of these genotypes in women with a normal cytology is an indication of increased risk of cervical cancer [21,37]. In a study published in 2004 on cervical cancer in Mozambique the dominant HPV types were 16 and 18, being present in 69% of tumours [38]. In a later study published in 2011 on HPV types in cervical cancers in Mozambique HPV 16 and 18 were the most common types detected in cervical cancer biopsies (among both HIV-negative and HIV-positive women) [39,40].

A high prevalence of HPV, across all age groups, particularly in HIV-positive women was found in this cohort. This emphasises the need to introduce cervical cancer screening programs in this group of women who are at high risk of developing cervical cancer. The WHO established guidelines to consider routine screening and treatment of cervical pre-cancer and provided different strategies for a national screening program [41,42]. These important recommendations for the choice of primary screening test and management of positive tests were based on resource availability. The use of a strategy to screen with an HPV test and treat is preferred over either a strategy to screen with visual inspection with acetic acid (VIA) and treat or a strategy to screen with cytology followed by colposcopy (with or without biopsy) and treat. Another possibility is to screen with an HPV test followed by VIA and treat [38]. Unfortunately, in Mozambique, there are lack of well-established routine cervical cancer screening programs. However, in a pilot program using care HPV as the screening test, it was demonstrated that HPV screening and treatment would be feasible in Mozambique.

Universal access to vaccination is the key to avoid most cases of HPV-attributable cancers [41]. Although our cohort does not study HPV in cervical cancer it does give information of the HPV types circulating in this population. All HPV genotypes were found with HPV 16, 18, 35, 52 and 68 being the most prevalent HR-HPV genotypes. None of the present vaccines target HPV 35. These genotypes are also not fully targeted by the bivalent and quadrivalent vaccines which targets only the genotypes 6, 11, 16 and 18. The relative contributions of HPV16 and 18 together and HPV 6, 11, 16, 18, 31, 33, 45, 52, 58 are 73% and 90%, respectively [39]. Thus, the nanovalent vaccine which targets the genotypes 6, 11, 16, 18, 31, 33, 45, 52, and 58, would be the recommended one. However, the burden of HPV16/18 in cervical cancers and the possibility of cross-protection with the bivalent and quadrivalent vaccines emphasizes the importance of the introduction of the more affordable vaccines in less developed countries [42], which is the case of Mozambique.

The study has limitations due to the lack of data available on cytology for the participants. In the HIV positive participants, we lacked data on anti-retroviral treatment.

## 5. Conclusions

This study found a high prevalence of HPV in this cohort. Across all age groups, multiple HPV infection was higher in HIV-positive women compared to HIV-negative women. High-risk genotypes were the most commonly found in the study cohort. Stratifications in age and HIV status are advantageous in studying the prevalence of HPV in a cohort of sexually active women to define target populations for cervical cancer screening based on HPV and plan implementation of such programs. Overall, the study results highlight the need to intensify sexually transmitted infection prevention interventions but also the need to implement HPV vaccination and cervical screening programs, as recommended in the millennium development goals to reduce cervical cancer by 2030 [4].

## Figures and Tables

**Figure 1 viruses-14-00024-f001:**
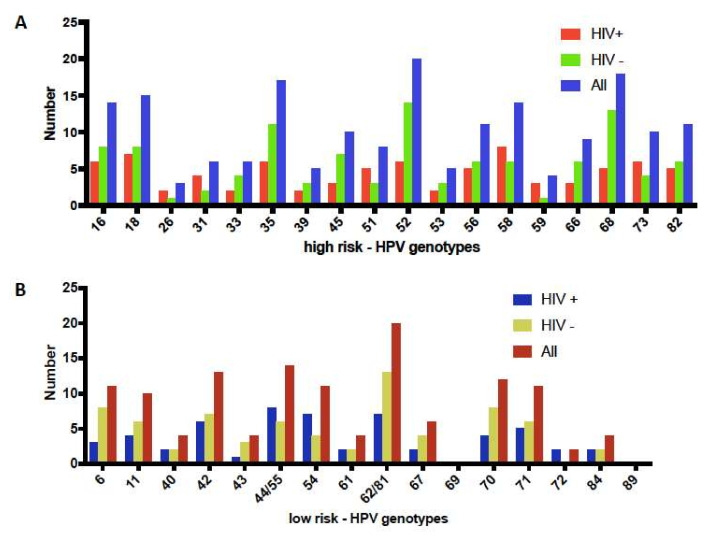
Number of women with each high-risk HPV genotypes (**A**) and low-risk genotypes (**B**) according to HIV status in the overall sample.

**Figure 2 viruses-14-00024-f002:**
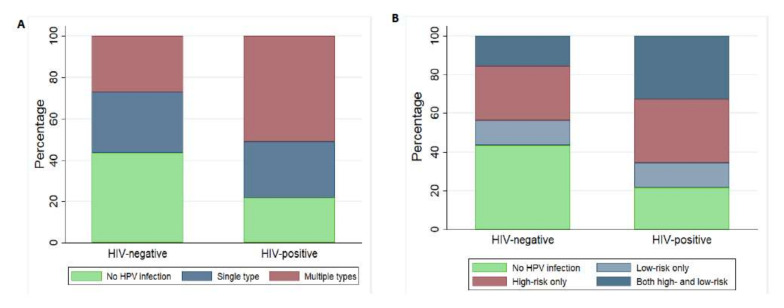
Percentage of women with single versus multiple types of HPV infections (**A**) and high and low risk categories HPV infections (**B**) according to HIV status.

**Figure 3 viruses-14-00024-f003:**
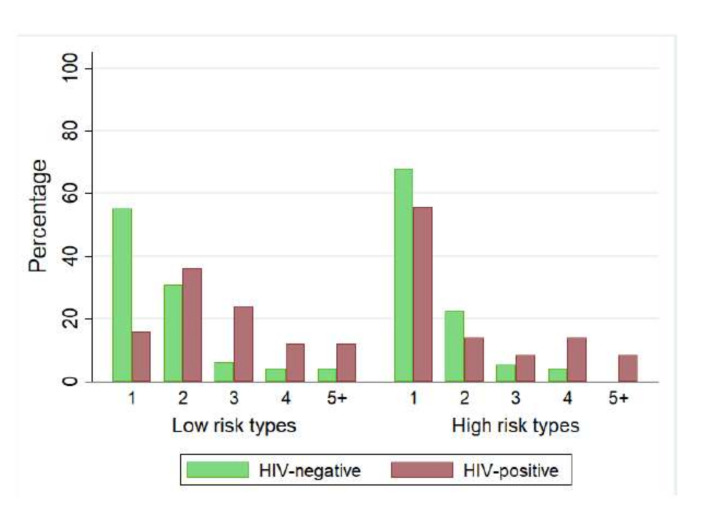
Distribution of multiple HPV infections in low and high-risk genotypes according to HIV status.

**Table 1 viruses-14-00024-t001:** Demographic and clinical data of study participants, by human immunodeficiency (HIV) status.

	HIV-Negative Women, N = 177	HIV-Positive Women, N = 56
Median age (range) 14–25 years 26–35 years 36–45 years >45 years	22 years (14–58 years) N = 121, median: 20 years N = 26, median: 30 years N = 22, median: 40 years N = 8, median: 48 years	34 years (19–62 years) N = 10, median: 23 years N = 22, median: 31 years N = 20, median: 39 years N = 4, median: 50 years
Age at first sex <16 years 16–18 years >18 years Missing	35 (19.8%) 110 (62.2%) 26 (14.7%) 6 (3.4%)	7 (12.5%) 31 (55.4%) 10 (17.9%) 8 (14.3%)
New sexual partners during past 3 months 0 1 2 3	153 (86.4%) 10 (5.7%) 4 (2.3%) 10 (5.7%)	47 (83.9%) 1 (1.8%) 3 (5.4%) 5 (8.9%)
Condom use during past 3 months No Sometimes Yes	91 (51.4%) 25 (14.1%) 61 (34.5%)	22 (39.3%) 8 (14.3%) 26 (46.4%)
Vaginal discharge	162 (91.5%)	48 (85.7%)
Vaginal ulcer	78 (44.1%)	32 (57.1%)
Cervical inflammation	85 (48.0%)	33 (58.9%)

**Table 2 viruses-14-00024-t002:** Human papillomavirus (HPV) by age and human immunodeficiency (HIV) status.

	All Women (N = 233)	HIV-Negative Women (N = 177)	HIV-Positive Women (N = 56)	*p*-Value
Any HPV All women 14–25 years 26–35 years 36–45 years >45 years	62.7% (146/233) 63.4% (83/131) 70.8% (34/48) 52.4% (22/42) 58.3% (7/12)	57.6% (102/177) 61.2% (74/121) 69.2% (18/26) 27.3% (6/22) 50.0% (4/8)	78.6% (44/56) 90.0% (9/10) 72.7% (16/22) 80.0% (16/20) 75.0% (3/4)	0.005 0.092 0.791 0.001 0.576
Multiple HPV infections All women 14–25 years 26–35 years 36–45 years >45 years	32.2% (75/233) 31.3% (41/131) 41.7% (20/48) 26.2% (11/42) 25.0% (3/12)	26.6% (47/177) 28.1% (34/121) 38.5% (10/26) 4.6% (1/22) 25.0% (2/8)	50.0% (28/56) 70.0% (7.10) 45.5% (10/22) 50.0% (10/20) 25.0% (1/4)	0.001 0.011 0.624 0.001 1.000
Single HPV infection All women 14–25 years 26–35 years 36–45 years >45 years	27.9% (65/233) 29.8% (39/131) 25.0% (12/48) 23.8% (10/42) 33.3% (4/12)	28.3% (50/177) 30.6% (37/121) 26.9% (7/26) 18.2% (4/22) 25.0% (2/8)	26.8% (15/56) 20.0% (2/10) 22.7% (5/22) 30.0% (6/20) 50.0% (2/4)	0.832 0.722 1.000 0.477 0.547
HR-HPV only All women 14–25 years 26–35 years 36–45 years >45 years	28.3% (66/233) 28.2% (37/131) 31.3% (15/48) 23.8% (10/42) 33.3% (4/12)	27.1% (48/177) 28.1% (34/121) 34.6% (9/26) 18.2% (4/22) 12.5% (1/8)	32.1% (18/56) 30.0% (3/10) 27.3% (6/22) 30.0% (6/20) 75.0% (3/4)	0.467 1.000 0.584 0.477 0.067
LR-HPV only All women 14–25 years 26–35 years 36–45 years >45 years	12.5% (29/233) 15.3% (20/131) 8.3% (4/48) 9.5% (4/42) 8.3% (1/12)	12.4% (22/177) 14.9% (18/121) 11.5% (3/26) 0.0% (0/22) 12.5% (1/8)	12.5% (7/56) 20.0% (2/10) 4.6% (1/22) 20.0% (4/20) 0.0% (0/4)	0.989 0.650 0.614 0.043 1.000
Both HR and LR-HPV All women 14–25 years 26–35 years 36–45 years >45 years	19.3% (45/233) 17.6% (23/131) 27.1% (13/48) 16.7% (7/42) 16.7% (2/12)	15.3% (27/177) 15.7% (19/121) 19.2% (5/26) 4.6% (1/22) 25.0% (2/8)	32.1% (18/56) 40.0% (4/10) 36.4% (8/22) 30.0% (6/20) 0.0% (0/4)	0.005 0.074 0.210 0.041 0.515

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
