# Peer review of "Identification of the Human Papillomavirus Genotypes, According to the Human Immunodeficiency Virus Status in a Cohort of Women from Maputo, Mozambique"

_viruses, 2021, doi:10.3390/v14010024_

Round 1

Reviewer 1 Report

This manuscript describes HPV type prevalence in a cohort of women in Maputo Mozambique, where 177 were HIV negative and 56 were HIV positive. The study sample was small, but the study well conducted and the data were informative, although not completely unexpected. More specifically, there was an overall high prevalence of HPV, and more so and in older women in the HIV infected group, although the latter conclusions need to be taken with some caution, since there were few older women in the HIV positive group.

In general the manuscript was well written and very easy to follow.

I have two comments.

  1. The authors write that the women included did not receive antimicrobial medications. Does this also include that they do not receive anti-HIV medication? This could be clarified in the material and methods section. (In the discussion the authors do namely write that they could not analyse for CD4+ cells, so that is fine).
  2. There is a typo on line 70, that should read in.. instead of In.

Author Response

1. The authors write that the women included did not receive antimicrobial medications. Does this also include that they do not receive anti-HIV medication? This could be clarified in the material and methods section. (In the discussion the authors do namely write that they could not analyse for CD4+ cells, so that is fine). 

Answer: We agreed with the reviewer’s suggestion and we have added in the material and methods section a text stating the exception of antiretroviral medications (line 103 and 104). Thank you.

2. There is a typo on line 70, that should read in.. instead of In.

Answer:We have corrected the typo on line 70. It now read “in”. (line 71). Thank you.

Reviewer 2 Report

This original article reports the prevalence of HPV infection stratified accordingly to HIV positivity and age of patients recruited in a health center in Mozambique.

Findings reported here are novel and interesting, since authors provide convincing evidence of the infection status in a poor studied population. In addition, these results could be useful also to extrapolate general features of infected patients that could be beneficial in other populations.

The study is well conceived, the manuscript is well written, and most of the figures and tables are easy to follow and self-explanatory.

Here a list of few minor suggestions to improve the clarity of the article:

  • Is very interesting that in the studied population, as well as in other patients from close regions, HPV16 is not the most prevalent HPV genotype, differently from worldwide studies. Please discuss more deeply the possible reasons of this finding.
  • In figure 2A-B, are these differences statistically significant? I surmise that the Chi-square test could help to identify p values of these results.
  • Figure 3 appears unnecessary and could be removed without compromising other findings

Author Response

1. Is very interesting that in the studied population, as well as in other patients from close regions, HPV16 is not the most prevalent HPV genotype, differently from worldwide studies. Please discuss more deeply the possible reasons of this finding.

Answer: We have added a discussion paragraph on the possible reasons for low prevalence of HPV-16 as you have suggested (line 296). Thank you.

2. In figure 2A-B, are these differences statistically significant? I surmise that the Chi-square test could help to identify p values of these results.

Answer: The statistical significance difference between the groups in figure 2A-B, was shown in table 2. Thus, we considered not to include this information in the figure.  Thank you.

3. Figure 3 appears unnecessary and could be removed without compromising other findings. 

Answer: While we do not refute the reviewers comment, we decided to maintain figure 3 because it better shows  how single and multiple infections in low and high risk genotypes are distributed among our study cohort groups. We consider it important since it can be facilitate choosing the correct vaccines for national vaccination programs. Thank you. 

Reviewer 3 Report

The manuscript by Cremildo Maueia et al. describes the distribution of the HPV genotypes in a cohort of 233 women from Maputo area. This is a descriptive study whereby the authors performed HPV genotyping of cervical swabs using a commercially available kit. Their major goal is to distinguish HIVpos versus HIVneg women. As the authors state in the discussion, many other reports have already demonstrated higher prevalence of HPV infections, mostly multiple infections, in HIVpos versus HIVneg. Unfortunately, they do not mention whether any cytological alterations were present in those women. With regard to the healthy status of the women, they have been classified using the following groups: vaginal discharge, vaginal ulcer or cervical inflammation. None of these definitions really fit with HPV-induced disease. The authors should make efforts to better classify the patients based on the presence or absence of any HPV-related disease as determined by visual inspection e.g. genital warts or cytology e.g. cytological abnormalities in the pap smears. I think this is the major flaw of this study. The overall prevalence of high-risk genotype infections is very high and this is very likely because the cohort is suffering from the bias that many women actually displayed HPV-related disease…they are not healthy…this is not clear…..The discussion is too long and then their vision is too much related to authors’ country. I understand they do not have a screening program in their country yet, but this has been already established since many years in many other countries…so they should tune down their conclusion because this is not new. Especially for HIVpos women, it is well known they are highly susceptible to HPV infection and related diseases. They should also mention that their study has some limitations as they can only detect the genotypes recognized by the kit, all the others are lost. Also the title does not properly describe what they have performed…...the term characteristics does not sound very well…they have not performed any characterization, they simply identify which genotypes are  present using a commercial kit that allows detection of a limited number of genotypes…please modify the title...  

Author Response

Answer: As we have described,  this study was conducted on a women seeking care regarding gynaecological symptoms and not on women with cervical cancer or seeking cervical cancer screening. There is no cervical cancer screening program in this population and so screening is not done.  This is the reason it is important to have information on HPV prevalence.

The paper intentionally emphasizes the local situation in Mozambique as the aim is to influence the public health policies of the country and to encourage the introduction of cervical screening based on HPV to meet the WHO targets.  It also feeds information into the HPV vaccine policy.

We considered to modified the title study to “Identification of the Human Papillomavirus  genotypes according to the Human Immunodeficiency Virus status in a cohort of women from Maputo, Mozambique”, as suggested.

We added as a study limitations the fact of not included information on cytology in the  cohort which could have interfered in the finding as well as the fact of our study had detected the genotypes recognized by the kit (line 327). Thank you.

Round 2

Reviewer 3 Report

The authors have addressed all the concerns raised during the first revision.